# Global floating algae blooms are expanding

Lin Qi [1,2], Menghua Wang [1], Brian B. Barnes [3], Douglas G. Capone [4], Joaquim I. Goes [5], Edward J. Carpenter[6], Yuyuan Xie[3] & Chuanmin Hu [3] ✉

In the past two decades, both microscopic algae (i.e., phytoplankton) and larger algae (i.e., macroalgae) have increased in certain coastal and open ocean waters, yet a comprehensive picture at the global scale is lacking. Here, we address this challenge by analyzing 1.2 million satellite images with computer artificial intelligence to quantify macroalgal mats and microalgal scums in global oceans between 2003 and 2022. With a total cumulative realized niche area of 43.8 million km², macroalgae blooms in the tropical Atlantic and western Pacific both expanded at unprecedented rates since the 2010s (13.4% per year since 2003). Although slower, the annual expansion rate of microalgae scums is also statistically significant (1.0% per year since 2003). Such trends are likely a result of ocean warming and eutrophication, with a possible regime shift favoring macroalgae and specialized species of microalgae. These findings have broad implications on ocean ecology, carbon sequestration, environments, and economy.

Most ocean waters are replete with microscopic algae (a.k.a. phytoplankton) and mixoplankton whose numbers can at times exceed millions of unicellular cells in a liter of water. Although many of these are small and invisible to the human eye, they supply roughly half of the consumable carbon and oxygen through photosynthesis[1] (with the other half generated by land vegetation). With modern satellite, shipboard, and laboratory techniques, their distributions, temporal changes, and responses to climate variability have been studied extensively[2,3]. In contrast, such information for larger algae or macroalgae (a.k.a. seaweed) that are clearly visible to the human eye is relatively scant. Similar to microalgae, macroalgae also serve important functions to support marine biota and maintain healthy ecosystems[4], yet excessive amounts of macroalgae in coastal waters and on beaches can be harmful to the environment, human beings, and economies[5,6]. Several reports showed recent expansions of the *Sargassum* and *Ulva* macroalgae in the North Atlantic Ocean and western Pacific Ocean, respectively[7–10]. For the reported macroalgae, these regions of increases coincide with those that take <20 years for climate change effects to emerge in ecological indicators as indicated by ocean color trends (Fig. 2b in Cael et al.[11]). Then, is there a regime shift in global macroalgae that showed dramatic changes in either distributions or abundance in the past 20 years? Where are they in the global oceans anyway?

Similar questions can be posed for certain types of microalgae as they can either adjust their buoyancy (due to the presence of gas vesicles) or swim using their flagella to form dense surface scums that often appear like foam, mats, or paint-like slicks[12]. These include *Trichodesmium spp.* that contributes approximately 50% of the nitrogen (N) fixation in global oligotrophic oceans[13] and *Noctiluca*, a species that is capable of forming extensive blooms in both coastal and open ocean waters[14]. While reports show increases in green *Noctiluca scintillans* in the northern Indian Ocean[15], global occurrences of surface microalgal blooms are largely unknown. Indeed, several studies of global *Trichodesmium* distributions report inconsistent and sometimes contradicting results[16–18].

Here, using 1.2 million satellite images and algorithms based on computer artificial intelligence and imaging spectroscopy, we provide a comprehensive picture of both types of floating algae (macroalgae mats and microalgae scums) in global oceans. Through analyzing the 20-year global maps of their distributions of abundance, we address the questions of where and when they can be found, what they are, how they have changed in the past 20 years, and what may happen in the future. We show that both macroalgae (*Sargassum*, *Ulva*) blooms and microalgae scums (*Trichodesmium*, *Noctiluca*, other cyanobacteria and dinoflagellates) are widely distributed in the Pacific Ocean, Atlantic

[1]NOAA Center for Satellite Applications and Research, College Park, MD, USA. [2]Global Science & Technology Inc., Greenbelt, MD, USA. [3]College of Marine Science, University of South Florida, St. Petersburg, FL, USA. [4]University of Southern California, Los Angeles, CA, USA. [5]Lamont Doherty Earth Observatory, Columbia University, Palisades, NY, USA. [6]Estuary & Ocean Science Center, Biology Department, San Francisco State University, Tiburon, CA, USA. ✉e-mail: huc@usf.edu

Ocean, Indian Ocean, and marginal seas such as the Baltic Sea, with a total cumulative realized niche area (i.e., geographic distribution area) of 43.8 million km² in the past 20 years. Of these, macroalgae have expanded at unprecedented rates since the 2010s, while the annual expansion rate of microalgae scums is also statistically significant, both due possibly to ocean warming and eutrophication. Based on these findings, we hypothesize that there may be a regime shift in global oceans to favor macroalgae.

## Results and discussion

### Where?

For convenience, we use FA to represent floating algae in this context, defined as either macroalgae mats on the ocean surface or microalgae surface scums. The methods to map and quantify them at the global scale can be found in Supplementary Figs. 1–5, with accuracy assessment presented in Supplementary Table 1.

Figure 1A shows a global map of mean FA density between 2003 and 2022, while their climatological monthly means and annual means are presented in Supplementary Figs. 6 and 7. The FA are found mostly in tropical and subtropical waters between 40°N and 40°S, with a total niche area of 43.8 million km², which is roughly 12% of the global ocean's surface area. They are found in most major oceans, from coastal to open waters (Fig. 1A inset). On the other hand, no FA were detected in the Arctic and Antarctic Oceans from this analysis.

In the Atlantic Ocean, in a water area of 19.8 million km², FA were mostly found in subtropical and tropical waters spanning from west Africa to the Caribbean Sea and Gulf of America (a.k.a. Gulf of Mexico). In addition, FA were also found in the Sargasso Sea and along the coast of SW Africa and SE America. A high-latitude marginal sea, the Baltic Sea, was the only identified FA-endemic region that is entirely above 40° latitude.

In the Indian Ocean, FA were found in waters with a surface area of 11.3 million km². These waters were mostly in the Arabian Sea and around Madagascar in the South Indian Ocean, in the Red Sea and Bay of Bengal, Southeast Asia, and the western coastal areas of Australia.

In the Pacific Ocean, most FA were found in the South Pacific, including waters around eastern Australia and islands of Fiji, Samoa, and Tonga. Additional FA were observed in the marginal seas bordering China, around Hawaii, and in coastal waters off North and South America. The total surface area of these waters was 12.7 million km².

### What are they?

By examining the spectral shapes of the identified FA and comparing them against a spectral library of known spectral shapes[19], the FA types in each location were inferred. The global FA-containing waters were divided into 14 zones (Fig. 1A, Table 1), with each zone being dominated by one or two types of FA. These FA were broadly categorized as macroalgae or seaweed (pelagic *Sargassum fluitans/natans*, *Ulva prolifera*, and floating *Sargassum horneri*) and microalgae scums (*Trichodesmium*, *Noctiluca*, other cyanobacteria and dinoflagellates) (Table 1), with their general distributions presented in Fig. 1B. Each type of FA has its own preferred environments including water temperature (Table 1). Most FA were found to peak in the spring and summer months (in both hemispheres) when water temperature is high and nutrients are relatively abundant.

Macroalgae were found in 2 major zones: Zone 3 in the North Atlantic and Zone 11 in the western Pacific, dominated by *S. fluitans/natans*[20] (Zone 3), and *U. prolifera* and *S. horneri*[10] (Zone 11), respectively, all peaked in spring and summer months. The 20-year map in Fig. 1 shows a water area of 17.4 million km² containing *S. fluitans/natans*, representing the biggest macroalgal bloom in global oceans as a result of the recent macroalgae expansion to the Great Atlantic *Sargassum* Belt[9]. With *U. prolifera* and *S. horneri* in Zone 11 contributing a cumulative footprint of 0.6 million km², total coverage of macroalgae can reach 18.0 million km², accounting for 41.1% of the FA-containing global waters (Table 1, Fig. 1B).

Except for Zone 3 and Zone 11, all other zones were found to be dominated by microalgae scums (Table 1), mainly from four types: *Trichodesmium spp.*, green *Noctilica*, dinoflagellates, and cyanobacteria (Fig. 1B).

*Trichodesmium* were found to have expanded in the global tropical and subtropical oceans, with a cumulative surface area of 20.6 million km². Other than the color coded regions in Fig. 1B, scattered *Trichodesmium* scums were also found in the Gulf of Mexico and along the Gulf Stream and Kuroshio Current, but they were negligible compared to the above regions.

The highest density of *Trichodesmium* was found in Zone 12 (4.4 million km²), followed by Zone 13 and Zone 8, which mostly peaked in February when water temperature was between 26–29°C (Fig. 1B inset). Accounting for >60% of global coverage, these regions have also been predicted, through biogeochemical modeling, to be the most likely areas for *Trichodesmium* occurrence[18]. Additionally, Zone

**A Average density of floating algae between 2003 and 2022**

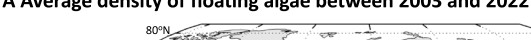

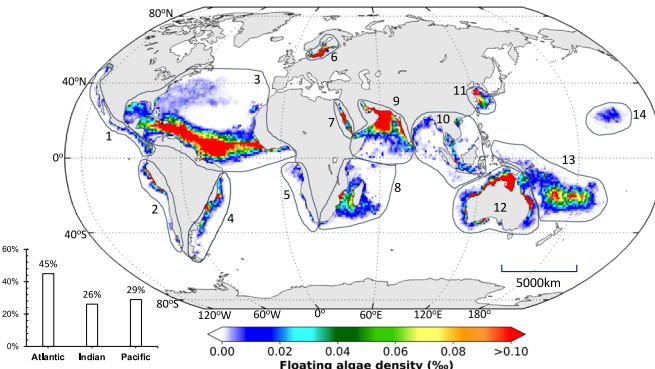

**B Distribution of floating algae types**

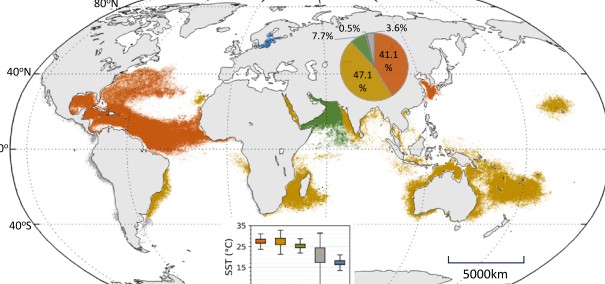

**Fig. 1 | FA density distributions and types in the global oceans. A** Mean FA areal density determined from MODIS observations between 2003 and 2022, where their surface areas are partitioned to the three major oceans (inset). Based on their locations and dominant types, they are grouped in the annotated 14 zones, with dominant type(s) listed in Table 1. Their climatological monthly means and annual means are presented in Supplementary Figs. 6 and 7. **B** Distributions of dominant FA types, determined from analysis of OLCI spectral shapes corresponding to (**A**).

Inset box plots show the quartile distribution of sea surface temperature (SST) for each FA type. The central line represents the median, the box boundaries indicate the 25th and 75th percentiles (interquartile range), and the whiskers denote the minimum and maximum values within 1.5× the interquartile range. Basemap generated using MATLAB M_Map package (www.eoas.ubc.ca/~rich/map.html). Source data are provided in the data repository[63].

**Table 1 | FA characteristics in the 14 zones shown in Fig. 1A**

| Zone # | Region | Dominate types | SST range | Peak month | Niche area (million km$^2$) | Proportion of global total |
|---|---|---|---|---|---|---|
| Zone 1 | Off northwest America | Dinoflagellates | 28–30°C | Feb | 0.6 | 1.4% |
| Zone 2 | Off southwest America | Dinoflagellates | 19–22°C | Mar | 0.6 | 1.5% |
| Zone 3 | North Atlantic Ocean | *S. fluitans/natans*; *Trichodesmium* | 27–29°C; 24–25°C | Jun; Sep | 17.4; 0.3 | 39.8%; 0.7% |
| Zone 4 | Off southeast America | *Trichodesmium*; Dinoflagellates | 24–27°C; 9–13°C | Nov; Sep | 1.2; 0.2 | 2.6%; 0.5% |
| Zone 5 | Off southwest Africa | *Trichodesmium*; Dinoflagellates | 27–28°C; 15–18°C | Apr | 0.3; 0.1 | 0.8%; 0.3% |
| Zone 6 | Baltic Sea | Cyanobacteria | 16–18°C | Jul | 0.2 | 0.5% |
| Zone 7 | Red Sea | *Trichodesmium* | 27–30°C | Jun | 0.3 | 0.8% |
| Zone 8 | Madagascar | *Trichodesmium* | 27–29°C | Jan | 3.6 | 8.3% |
| Zone 9 | Arabian Sea | Green *Noctiluca*; *Trichodesmium* | 24–26°C; 29–30°C | Feb; May | 3.4; 0.8 | 7.7%; 1.8% |
| Zone 10 | Southeast Asia | *Trichodesmium* | 28–30°C | Mar | 1.7 | 3.8% |
| Zone 11 | Marginal Seas off China | *S. honeri*; *U. prolifera* | 15–19°C; 19–25°C | Apr; Jun | 0.5; 0.1 | 1.1%; 0.2% |
| Zone 12 | Australia | *Trichodesmium* | 26–29°C | Nov | 4.4 | 10.1% |
| Zone 13 | Southwest Pacific | *Trichodesmium* | 26–28°C | Feb | 6.7 | 15.3% |
| Zone 14 | Hawaii | *Trichodesmium* | 26–27°C | Sep | 1.3 | 2.9% |

The SST range represents the quartiles of SST for each zone. In some zones, two types of FA were observed, where the SST range and peak month are listed separately for each type. Source data are provided in the data repositories[62,63].

14 around Hawaii is the only location in an ocean gyre with recurrent *Trichodesmium* blooms. Globally, the distributions of the *Trichodesmium* scums are generally in line with the required oceanic and atmospheric conditions (Supplementary Fig. 8) except the region around New Caledonia. We speculate that the frequent volcano eruptions in this region may supply iron to stimulate *Trichodesmium* growth[21].

Green *Noctiluca* is a heterotroph-turned-mixoplankon (a.k.a. mixotroph)[22] that can form intense blooms in both coastal and open ocean waters[23,24]. Green *Noctiluca* is capable of movement with its flagella, and the increase in lipids and ammonium within their cells makes them highly buoyant and easily prone to dispersal by surface currents. Green *Noctiluca* scums were most evident in Sea of Oman and the Arabian Sea, occupying a cumulative surface area of 3.4 million km². The scums were usually found during late winter and spring (Supplementary Fig. 6) when water temperature was between 24–26°C (Table 1) and the water was highly stratified to meet the *Noctiluca*'s metabolic needs over other phytoplankton such as diatoms[25].

Other dinoflagellate surface scums were found in coastal waters off the west coast of the Pacific Ocean, the east coast off South America, and the west coast off South Africa. These represent various types of dinoflagellates[26]. Collectively, these dinoflagellate surface scums occupy a water area of 1.6 million km².

Lastly, surface scums of various types of cyanobacteria (e.g., *Nodularia spumigena* and *Aphanizomenon*) were often found in the Baltic Sea. These cyanobacterial blooms have been widely reported in the past[27,28], and play important roles in regional ecology and biogeochemistry. The blooms are strictly a summer phenomenon during the brief period of upper water column stratification and stability, where *Aphanizomenon spp*. usually initiate the bloom in early summer which are succeeded by *Nodularia spumigena*[28].

## Long-term trends?

The decadal maps in Fig. 2A, B clearly reveal increased FA in the past 10 years relative to the first 10 years, with most increases found in the tropical Atlantic and western Pacific. The de-seasoned trend analysis further indicates that, when all microalgae and macroalgae are integrated as two classes, both microalgae and macroalgae showed statistically significant increasing trends (Fig. 2C, D). All 5 major types of FA (Fig. 1B inset) have increased significantly in the past ten years, with three of them (Seaweed, *Trichodesmium*, green *Noctiluca*) being statistically significant during the past 20 years. Of all 14 zones, 7 showed

statistically significant increasing trend, while only 2 showed statistically significant decreasing trend (Fig. 3).

While some of the increased microalgae scums are in line with a recent report on coastal phytoplankton blooms[3], of particular importance are the increases in *Trichodesmium* scums in both coastal and open oceans.

In all regions rich in *Trichodesmium* scums, there are multiple sources to supply iron, for example, dust and ash deposition around Australia and Arabian Sea, terrestrial discharge around Madagascar, and lava flows and submarine discharges (e.g., the Loihi seamount) around Hawaii[29]. In an iron replete warm ocean, the growth of *Trichodesmium* is often constrained by the critical nutrient phosphorus (P)[30]. Unlike N, P is mostly derived from terrestrial sources. Rockström et al. identified and quantified 10 planetary boundaries that should not be transgressed to avoid unacceptable environmental changes caused by human activities[31]. These boundaries have been revisited recently[32], where the transgressed boundaries increased from 3 to 6, with one of the newly transgressed boundaries being P at both global and regional scales. More transgressed boundaries suggest more environmental changes. As such, the increased P is possibly one major reason behind the increased *Trichodesmium*.

Green *Noctiluca* blooms in the Arabian Sea have been tied to the upshoaling of hypoxic waters into the euphotic zone[14,33], an apparent change in nutrient stoichiometries driven primarily by decline in inorganic nitrate, increased surface water warming, and stratification[15]. Unlike diatoms, green *Noctiluca* grows preferentially on urea and ammonium instead of nitrate and can meet its nitrogen requirements primarily from prey ingestion. Additional phosphorus and iron inputs from winter dust storms also stimulate the growth of green *Noctiluca*. These factors collectively determine the long-term increasing trends in the observed surface scums in the Arabian Sea (Fig. 3).

The rapid expansion of *U. prolifera* and *S. horneri* in the marginal seas of the western Pacific has been shown to be related to multiple factors including increased ocean temperature, eutrophication, and increased human activities (agriculture and aquaculture)[10,34]. The rapid expansion of the Atlantic *S. fluitans/natans*, on the other hand, is believed to be caused by a tipping point in 2010 due to climate variability, when long-distance transport from the Sargasso Sea provided initial *Sargassum* population to the tropical Atlantic where all conditions (light, temperature, nutrients) favor *Sargassum* growth[35]. The inter-annual changes are shown to be driven by physical forcing such as winds, circulation, and upwelling[36]. Diazogrophs found on the

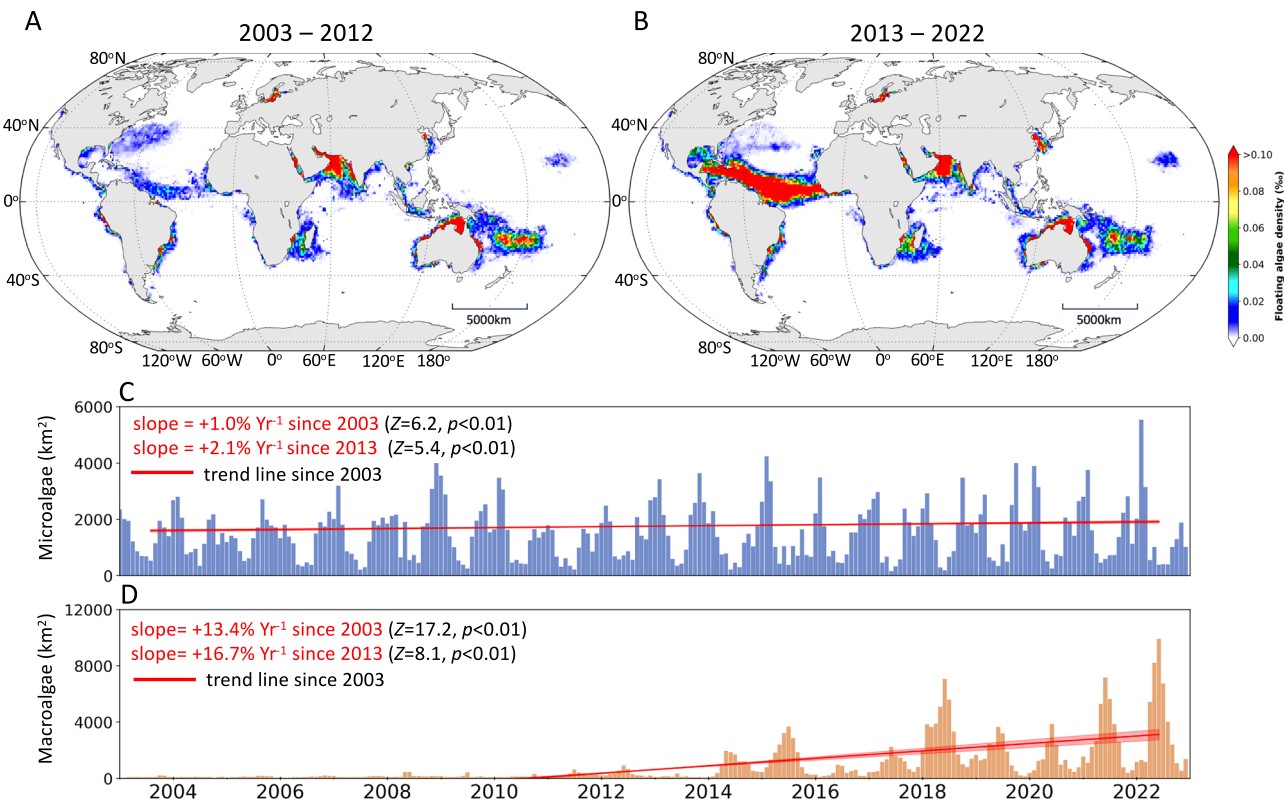

**Fig. 2 | Decadal changes in FA density distributions. A** FA density distributions between 2003–2012; **B** FA density distributions between 2013–2022. The 20-year changes in FA abundance are shown in (**C**) for microalgae scums and (**D**) for macroalgae mats, respectively. The year mark starts from January. Vertical lines represent the monthly means, and the shadowed red lines represent the trend lines of the de-seasoned data with 95% confidence intervals. Theil-Sen's slopes, p values and Z statistics determined from the two sided Mann-Kendall test[64] are annotated. Basemap generated using MATLAB M_Map package (www.eoas.ubc.ca/-rich/map.html). Source data are provided in the data repositories[62,63] as well as in a Source Data file.

epiphytes of both *S. fluitans/natans*[37,38] and *S. horneri*[39] can also provide new nitrogen to *Sargassum* growth. Furthermore, increased phosphorus in the tropical Atlantic, as revealed by long-term coral coring isotope analysis, could explain the recent increasing trend in the Atlantic *Sargassum*[40].

What is noteworthy is that most increases in both floating macroalgae and microalgae scums occurred in the recent decade (Supplementary Fig. 9), in line with the accelerated global ocean warming since 2010. In particular, the "tipping points" of macroalgae blooms were all around 2010: the first major *Ulva* bloom occurred in 2008[41], the first major *S. horneri* bloom occurred in 2012[10], and the first major *S. fluitans/natans* bloom occurred in 2011[7,9]. Are these near-synchronized blooms of three different types of macroalgae, far apart in two major oceans, simply due to coincidence, or a result of regime shift in oceanographic conditions that favor macroalgae growth?

Indeed, in the same location where *Sargassum* has thrived in recent years, phytoplankton chlorophyll pigment concentration (Chla, mg m$^{-3}$), derived from satellite observations, did not show increasing patterns in the western tropical Atlantic even under the Amazon's influence[42]. The same analysis was extended to the Yellow Sea and East China Sea where *U. prolifera* and *S. horneri* thrived in recent years, with similar results obtained: Chla showed no apparent increases (Supplementary Fig. 9). Such contrasting trends may be due to the fact that, unlike most microalgae, macroalgae face far less grazing pressure, thus are more responsive to environmental changes such as ocean warming and eutrophication. A recent report showed that this may be the case for the Yellow Sea and East China Sea, where a shift from microalgae to macroalgae has been observed from field surveys[43].

Extensive field data for phytoplankton community composition in the tropical Atlantic are not available, but the same reason might be applicable for the Atlantic *Sargassum*.

Thus, while changes in the planetary boundaries have been chronicled[31,32], their impacts on macroalgae and microalgae may be different. Once the boundaries are transgressed (e.g., phosphorus), continued increases may result in a tipping point that causes contrasting responses for macroalgae and microalgae. If this is the case, we believe that a regime shift in oceanographic conditions has already occurred to favor macroalgae, which will have profound impacts on radiative forcing in the atmosphere[44] and light availability in the ocean, as well as on carbon sequestration, ocean biogeochemistry, and upper ocean stability[45–48].

Our study shows that global oceans are changing at an unprecedented pace, and the appearance of macroalgal mats and microalgal surface scums represent manifestations of this change. With continuously increasing temperature and nutrient inputs due to both natural and anthropogenic factors, there is every indication that different ocean basins will experience more transgressions of planetary boundaries in the not-so-distant future, which will strengthen the disparity between macroalgal and microalgal abundance. The recent occurrence of macroalgae blooms may constitute a regime shift that can have profound impacts on all aspects of ocean ecology, carbon sequestration, fisheries, tourism, and economy. While the study shows a comprehensive picture of floating macroalgae mats and microalgae scums at the global scale, understanding how they respond to environmental changes and especially the interplay between their own processes under the same changing oceanographic conditions is still a major task for the research community. One thing is nearly certain, though: as both ocean temperature and phosphorus flux are projected

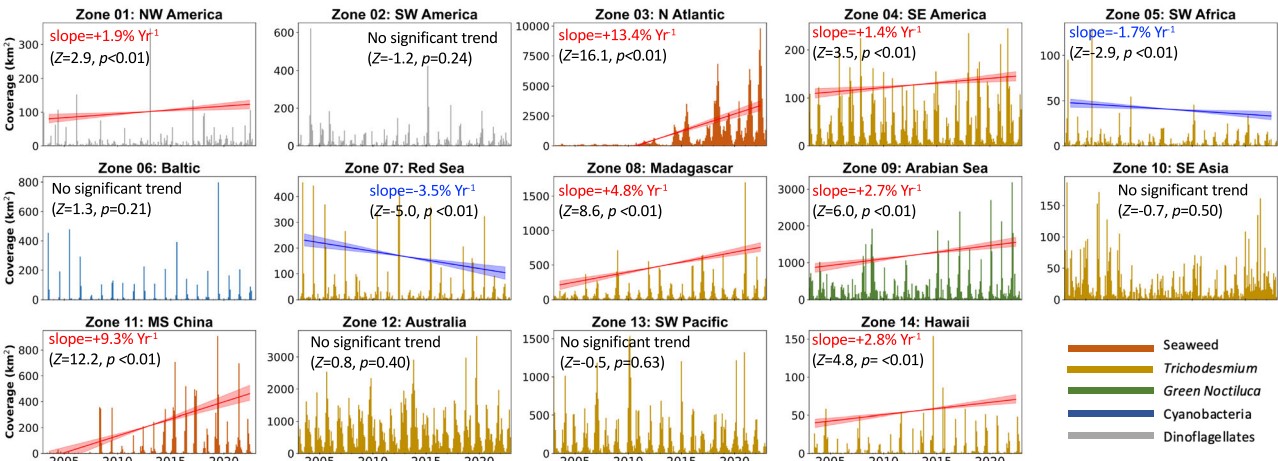

**Fig. 3 | Long-term changes in floating macroalgae mats and microalgae scums.** The changes are presented for each of the 14 zones outlined in Fig. 1A. The year mark starts from January. Vertical bars represent monthly means color coded by the algae type, and the shadowed red or blue lines represent the trend lines of the de-seasoned data with 95% confidence intervals. Theil-Sen's slopes, *p*-values and *Z* statistics determined from the two sided Mann-Kendall test[64] are annotated. Both red and blue lines represent statistically significant trends. Note the scale difference on the y-axis. Source data are provided in a Source Data file.

to rise, both macroalgae and *Trichodesmium* are likely to continue to expand in the near future.

## Methods

### Data sources

**Satellite data.** MODIS/Aqua Level-1A data from 2003 to 2022 were downloaded from NASA OB.DAAC (https://oceancolor.gsfc.nasa.gov) and processed to Rayleigh-corrected reflectance ($R_{rc}(\lambda)$, dimensionless) for each spectral band at 1-km resolution using the software SeaDAS (version 8.2). The use of $R_{rc}(\lambda)$ data instead of fully-corrected surface reflectance in all subsequent analyses is because the latter would depend on the assumption of zero water-leaving reflectance contributions at the near-infrared (NIR) and/or shortwave infrared (SWIR) bands. Such an assumption often fails over macroalgae mats or microalgae scums because these surface plants have elevated NIR and SWIR reflectance[49]. For the entire globe of 90°N–90°S and 180°E–180°W, a total of 1.2 million MODIS Level-1A 5-min granules were processed. These data were used to develop the deep-learning algorithm to extract and quantify FA features, and to establish time-series maps. In future works when MODIS stops functioning due to aging, VIIRS satellite sensors can also be used in a similar fashion to map global floating algae[50].

OLCI/Sentinel-3A and -3B data (300 m resolution) were also downloaded from NASA OB.DAAC and processed in the same away as with MODIS. Because OLCI sensors have more spectral bands, the OLCI $R_{rc}(\lambda)$ were used for spectral analysis to distinguish the FA types[19]. OLCI data were used exclusively for FA type differentiation and not for FA mapping, and therefore not projected to the same resolution as MODIS.

**Environmental data.** Sea surface temperature (SST). NOAA monthly Optimum Interpolation SST (OISST) data (4 km resolution), covering the period of 2003 to 2022, were downloaded from the Google Earth Engine (GEE) platform. These data were used to determine the SST range from the FA-containing waters. Details of how the OISST data products were generated can be found in ref. 51.

Dust and black carbon. Monthly mean column mass density of dust (dust, mg m$^{-2}$) and black carbon (BC, mg m$^{-2}$) were downloaded from the GEE platform for the period of 2003–2022. These gridded data (0.5° × 0.625° grid) were generated using the second Modern-Era Retrospective analysis for Research and Applications (M2T1NXAER) model and provided by the U.S. NASA.

Sea surface nutrient. Multi-year average cumulative sea surface nutrient concentration data for each climatological month, including nitrate (N) and phosphorus (P), were obtained from NOAA WOA18. These gridded data (1 × 1° grid) were used to compute the global distributions of N/P ratios.

These Environmental data were interpolated using the nearest-neighbor method to align with the monthly FA products for comparative analysis, where the methods used to derive the FA products are described below.

### Mapping and quantifying FA

The methods have been described in detail in several recent publications for both macroalgae mats[52] and microalgae scums[50]. For completeness, the steps are summarized in the flow chart of Supplementary Fig. 1 and described here. Basically, the methods include the use of a ResUNet deep-learning (DL) model to delineate FA image features automatically from MODIS images, a spectral diagnostic step to classify the FA type from the delineated features, a spectral unmixing to quantify FA density (% cover with a pixel) from the delineated features, and an image composition to generate global and regional FA maps.

**Model selection.** Among the various machine learning approaches, UNet as a convolutional neural network has shown superior performance over other traditional machine learning approaches (e.g., Multi-Layer Perception, Random Forest, Support Vector Machine, and Boosted Regression Trees) when extracting floating algae image features (Supplemental Materials of Wang and Hu[53]). Based on this finding, the segmentation deep-learning (DL) ResUNet model[54] has been adopted to extract floating algae pixels because of its ability of iterative training to minimize both false positive and false negative detections. The ResUNet-based DL models have therefore been developed and validated for macroalgae detection[52] and microalgae scum detection[50] using 1-km resolution satellite images where the image features are very weak because the FA density is mostly <1% of a pixel size. The principles and practices of detecting and quantifying floating algae have been reviewed by Hu et al.[55,56].

ResUNet is not the only choice for improved image segmentation. In recent years, Vision Transformers[57] (ViT) or other forms of deep-learning architectures[58] (e.g., discriminant superpixel graph or dSPG) have also shown superior performance over traditional machine learning approaches. These alternative approaches, however, have

been primarily used in object recognition on land[57] or water objects (e.g., oil slicks) in meter-resolution images[59]. In contrast, our target pixels are weak image features from water, with FA density mostly occupying <1% of an image pixel (1-km resolution). These weak image features make them very difficult to extract as compared to features on land or in high-resolution images. In earlier trail-and-error tests, the ViT's core component, namely "self attention", was incorporated in the ResUNet, but with unsatisfactory results: similar weak features due to cloud-adjacency effects, cloud shadows, or image noise were often falsely recognized as FA features. We therefore selected ResUNet architecture and optimized its performance through model iterations and proper parameterization. The models generally showed 90% or higher accuracy when being applied to detect FA pixels in 1-km resolution images[50,52,60]. Because of this, the ResUNet-based DL model has been implemented to monitor and track *Sargassum* macroalgae in the Atlantic Ocean in an operational fashion (i.e., with daily updates in near real-time) in the *Sargassum* Watch System (SaWS; https://optics.marine.usf.edu/projects/saws.html). Of course, this does not mean that ResUNet is the best architecture in DL models, but it has been proven effective with acceptable accuracy for the purpose of global mapping in this study[50,52,60]. In the near future we will continue our effort in finding the best approach, for example, by possibly combining the advantages of architectures of transformers and ResUNet, or adapting to the most recent progress in image segmentation techniques.

**DL model set up.** As described in Hu et al.[52], MODIS $R_{rc}$ bands (412, 443, 488, 547, 667, 748, and 869 nm) and one AFAI band were used for the DL model development. Here, AFAI is defined as[49,61]:

$$AFAI = R_{\lambda_2} - \left[ R_{\lambda_1} + \frac{\lambda_2 - \lambda_1}{\lambda_3 - \lambda_2} \times \left( R_{\lambda_3} - R_{\lambda_1} \right) \right], \qquad (1)$$

where the three MODIS wavelengths are $\lambda_1 = 667$ nm, $\lambda_2 = 748$ nm, and $\lambda_3 = 869$ nm. The index is a measure of the elevated reflectance at $\lambda_2$ (i.e., the red-edge reflectance) referenced against a linear baseline formed between $\lambda_1$ and $\lambda_3$. Such a linear design can be traced back to the original FAI[49], which has been shown to be more tolerant than other indexes to perturbations by aerosols, thin clouds, and moderate sun glint.

The structure of the ResUNet DL model is presented in Supplementary Fig. 2, with details presented below.

**Prepare "ground truth" labels for DL model.** A graphical user interface (GUI) developed in MATLAB was used to automatically generate "ground truth" images from AFAI images using dynamic thresholds tailored for each manually selected region of interest (Supplementary Fig. 3). For confusing features such as nearshore shallow waters, cloud edges, cloud shadows, and image noise, adjustments were made through the GUI to label these features as true negatives instead of true positives (Supplementary Fig. 3). This approach was used to prepare all "ground truth" images used for both model training and model validation.

As stated earlier, although the ResUNet architecture has been proven effective in delineating FA features and therefore selected for this work, it is unknown whether it is the best architecture. Therefore, we have made the complete "ground truth" dataset, including both training and validation data, publicly available through a data repository (https://doi.org/10.17632/f39zt9g2c4.1), thus enabling future studies to benchmark and compare with other methods.

**DL model parameterization.** The architecture of the ResUNet model is illustrated in Supplementary Fig. 2. The input to the DL model is an 8-layer image with a size of 256 × 256 pixels, consisting of 7 normalized

MODIS $R_{rc}$ bands and 1 AFAI band. The normalization procedure follows the approach described in Hu et al.[52] and Wang and Hu[53].

The MODIS L2 Images (image size: ~1300 × 2000 pixels) were divided into sub-images of 256 × 256 pixels for model training and applications. Edging sub-images smaller than this size were padded with Not-a-Number (NaN) values to ensure a consistent input size. After running the model, the sub-images were reassembled to reconstruct the full-size image output.

A total of 6912 sub-images were extracted from 830 MODIS L2 images spanning the global ocean to construct the training dataset. Rather than using all sub-images, only those containing typical FA features or potentially misleading patterns (e.g., clouds, cloud shadows, sun glint, or nearshore shallow water) were selected to improve model learning and reduce ambiguity. Among the selected sub-images, 70% were used for training and 30% for testing. The model was trained with a batch size of 32. The initial learning rate was set to 0.05 and updated according to Eq. 2 starting from epoch 11. The loss curve over training epochs is shown in Supplementary Fig. 3. The loss value steadily decreased as the number of epochs increased, and began to stabilize after approximately 30 epochs. Training was stopped at epoch 50 to avoid overfitting and maintain optimal performance.

$$lr_{new} = lr_{current} \times e^{-0.1} \qquad (2)$$

**DL model accuracy assessment.** To evaluate the DL model performance, MODIS images containing various FA features around the globe were selected. Due to the large volume of available data (1.2 million MODIS L2 images) and temporal variability in FA distributions, three representative years (2003, 2013, and 2022) were selected for model evaluation. Images from these years were randomly sampled to ensure coverage of a variety of oceanic and coastal environments, as well as a wide range of FA densities. FA features associated with macroalgae (*Ulva* and *Sargassum*) and microalgae (including *Trichodesmium*, green *Noctiluca*, dinoflagellates, and cyanobacteria) were evaluated separately to facilitate comparison with previous studies. A total of 15 L2 images containing macroalgae and 19 L2 images containing microalgae were selected for validation.

The "ground truth" images were generated using the same procedure describe above and shown in Supplementary Fig. 3. All image pairs, including the "ground truth" images and the corresponding DL model output images, are available at a public data repository[62] (https://doi.org/10.17632/f39zt9g2c4.1). The FA types present in each image are also documented under the same file folder.

The performance of the DL model was assessed using a set of commonly used statistical metrics (Eqs. 3–6), which provide a quantitative measure of the model's accuracy and reliability in detecting and segmenting FA features. The validation results are summarized in Supplementary Table 1.

$$Precision = \frac{TP}{TP + FP} \times 100\% \qquad (3)$$

$$Recall = \frac{TP}{TP + FN} \times 100\% \qquad (4)$$

$$F1 = \frac{2 \times Precision \times Recall}{Precision + Recall} \times 100\% \qquad (5)$$

$$IoU = \frac{TP}{TP + FP + FN} \times 100\% \qquad (6)$$

Here, true positive (TP), false positive (FP), and false negative (FN) values were used to assess model performance, representing correctly identified, incorrectly identified, and missed detections, respectively. The F1 score (Eq. 5) and Intersection over Union (IoU, Eq. 6) were

calculated to summarize model accuracy, incorporating both types of classification errors. To account for pixel-level uncertainty, a weighting method was implemented in which each pixel's influence on TP and FP was modulated by its corresponding χ value, where χ is estimated as the sub-pixel fractional cover of the FA (0–100%, Eq. 7). This weighting approach was adapted from Hu et al.[52] and Qi et al.[50] to improve robustness in performance evaluation.

As shown in Supplementary Table 1, the F1 score and IoU achieved in this study are similar to those reported for *Sargassum* detection in the Atlantic Ocean[52] and *Trichodesmium* detection around Australia[50]. The detection accuracy for macroalgae is slightly lower than for microalgae, likely due to the presence of small, low-density macroalgae patches that are difficult for the model to identify reliably. Nevertheless, given that the detection in this study covers a global scale and includes multiple types of FA, the accuracy is considered to be acceptable.

**Identify FA types.** With the ResUNet model segmentation results, the *FA* type of model-extracted FA features is determined through examining the spectral shapes of randomly selected *FA* pixels (Eq. 7) and comparing them against a spectral library of known *FA* types following the methods of[19].

$$
\begin{aligned}
\Delta R_{rc}(\lambda) &= R_{rc}^{T}(\lambda) - R_{rc}^{W}(\lambda) \\
&= [\chi R^{FA}(\lambda) + (1-\chi)R^{W}(\lambda)] - R^{W}(\lambda) \\
&= \chi(R^{FA}(\lambda) - R^{W}(\lambda)) \\
&\approx \chi R^{FA}(\lambda)\,[\text{assuming } R^{W}(\lambda) \ll R^{FA}(\lambda)]
\end{aligned}
\tag{7}
$$

Here, the superscripts "*T*" and "*W*" represent target pixel and nearby water pixel in the image, and $R^{FA}(\lambda)$ indicates the *FA* end-member reflectance spectrum. Here, χ represents the subpixel FA coverage within the target pixel (i.e, percent coverage). The transition from $R_{rc}$ to surface reflectance $R$ is based on the assumption that aerosol reflectance over the target pixel and nearby water pixel is the same, and therefore cancelled in the two $R_{rc}$ terms in the first row. From Eq. (7), the spectral shape in $\Delta R_{rc}(\lambda)$ is the same as in the end-member $R^{FA}(\lambda)$, thus can be used to infer the FA type.

**Quantify FA density and amount.** To estimate the FA's areal density (percent coverage, χ in Eq. 7), each FA pixel was spectrally unmixed using Eq. 8.

$$
\Delta AFAI = \chi(AFAI^{FA} - AFAI^{W}) \approx \chi AFAI^{FA}.
\tag{8}
$$

The approximation is based on the fact that $AFAI^{W} \approx 0$ for most waters unless they are very turbid. Then, χ is determined by the ratio between the image based $\Delta AFAI$ and endmember $AFAI^{FA}$ (a constant value that does not change in either space or time).

Once the individual L2 images are used to generate *FA* pixels with their types and subpixel areal density determined, monthly and annual means are generated using the SeaDAS l2bin and l3bin routines. Finally, the Level-3 data are projected to generate global distribution maps of *FA* abundance. Because such detected *FA* are either macro-algae mats or microalgae (i.e., phytoplankton) scums, certain types of microalgae that have a strong color signal but do not form surface scums (e.g., *Phaeocyctis*, *Coccolithophores*) are not detected by the DL model.

The processes of how the DL model was applied to extract FA pixels, how the FA type was determined through spectral diagnostics, and how the final FA areal density was determined through spectral unmixing is illustrated in Supplementary Fig. 5 for each type of FA.

Such derived monthly climatology of global distributions of FA density between 2003 and 2022 are shown in Supplementary Fig. 6,

while the annual distributions of FA density between 2003 and 2022 are shown in Supplementary Fig. 7. All global maps are made available through the figshare repository[63] (https://doi.org/10.6084/m9.figshare.28139492). To examine the environmental factors that may control the distributions of global *Trichodesmium*, the global distributions of several environmental variables that are important to *Trichodesmium* growth are shown in Supplementary Fig. 8, where the *Trichodesmium* boundaries determined from this study (Fig. 1B) are overlaid as black outlines. Finally, the long-term changing patterns of three types of macroalgae in their respective niche regions are presented in Supplementary Fig. 9, where long-term changes of their respective water-column Chla concentrations are also presented for comparison.

### Statistical analysis
For each identified FA type, once the 20-year time series of areal coverage is established, it is first de-seasoned (i.e., removing the seasonal variations) using a Python module. Then, the Mann-Kendall test was used to determine the long-term trend together with its statistical significance (*p* value) and the Thei-Sen estimator/slope and 95% confidence intervals. The Mann-Kendall test was provided through an open-source Python package[64].

### Reporting summary
Further information on research design is available in the Nature Portfolio Reporting Summary linked to this article.

## Data availability
All satellite data used in this analysis is available through the NASA OB.DAAC (https://oceancolor.gsfc.nasa.gov). The environmental data used in this analysis is available through their specific data providers (https://disc.gsfc.nasa.gov/datasets/M2T1NXAER_5.12.4/summary; https://www.ncei.noaa.gov/access/world-ocean-atlas-2018/bin/woa18oxnu.pl). The software used to process satellite data is SeaDAS (version 8.2, https://seadas.gsfc.nasa.gov). All satellite data used in the training and validation of the deep-learning model to detect floating algae from MODIS images has been made available through a public data repository[62] (https://doi.org/10.17632/f39zt9g2c4.1), including the spectral reflectance data, quick-look images, "ground truth" images, and model output images. The global floating algae maps used in this study are made available through the figshare repository[63] (https://doi.org/10.6084/m9.figshare.28139492). Source data are provided with this paper.

## Code availability
All computer codes to process satellite data are available through SeaDAS (version 8.2, https://seadas.gsfc.nasa.gov). The Python and MATLAB codes to train the DL model and analyze time-series data are available from the corresponding author upon request.

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

## Acknowledgements

This work was supported by the U.S. NASA (80NSSC20M0264 (C.H.), 80NSSC24K1507 (C.H.), 80NSSC25K7361 (B.B.B.), 80NSSC25K7427 (C.H.), 80NSSC24K0893 (J.I.G.), 80NSSC25K7239 (J.I.G.)), the U.S. NOAA (NA23NOS4780291 (B.B.B.)), the U.S. EPA (02D42923 (B.B.B.)), and the U.S. Joint Polar Satellite System (JPSS) program (ST133017CQ0050_1332KP22FNEED0042 (L.Q.) and ST13301CQ0050/1332KP22FNEED0042 (C.H.)). We thank the U.S. NASA OB.DAAC for providing all satellite data and thank the NASA OBPG for providing the SeaDAS software to enable this analysis. The scientific results and conclusions, as well as any views or opinions expressed herein, are those of the author(s) and do not necessarily reflect those of NOAA or the Department of Commerce.

## Author contributions

Conceptualization: C.H. and L.Q.; Methodology: L.Q. and C.H.; Investigation: L.Q. and C.H.; Visualization: L.Q., C.H., B.B.B., and M.W.; Writing – original draft: L.Q.; Writing – review and editing: L.Q., M.W., B.B.B., D.G.C., J.I.G., E.J.C., Y.X., and C.H.

## Competing interests

The authors declare no competing interests.
