## [Peer Review file · Nature Communications]

Global floating algae blooms are expanding

Corresponding Author: Professor Chuanmin Hu

Version 0:

Reviewer comments:

Reviewer #1

(Remarks to the Author)

I have completed the review of the manuscript titled "Global Floating Algae Are Rising," submitted for consideration in Nature Communications. This study presents a novel global analysis of floating algae (FA), using an extensive dataset of satellite imagery over a 20-year period. Including different species and associated environmental factors enhances its originality and significance. While the study contributes valuable insights into FA dynamics, certain issues must be addressed before the manuscript can be considered for publication. Below, I provide a detailed review of the article.

1. The study employs a deep-learning model (ResUNet) for FA detection but does not provide sufficient details about the training and validation datasets.

The accuracy and reliability of a machine learning model depend heavily on the quality and quantity of these datasets. The manuscript states: "The ResUNet deep-learning model is trained and validated using carefully prepared 'ground truth' images, with detailed steps described in [48]. For simplicity, such details are not repeated here." However, reference [48] mentions that the ground-truth images were labeled using a combination of manual outlining and objective delineation. Given the vast number of MODIS images analyzed in this study, how was ground-truth data manually determined at such a large scale? Additionally, what quality control measures were implemented to ensure accuracy? The authors should provide a detailed explanation of the ground-truth dataset, including its accuracy, potential biases, and whether any independent validation was performed.

2. The study employs the ResUNet deep-learning model but does not justify why this model was chosen over more advanced alternatives, such as Vision Transformers (ViT) or other state-of-the-art deep-learning architectures. A comparative analysis of model performance (e.g., accuracy, computational efficiency, robustness) would strengthen the manuscript's credibility. If ResUNet was selected due to specific advantages, these should be clearly outlined.

3. The manuscript does not provide sufficient details on the ResUNet model's structure. The authors should clearly describe the model architecture and parameter selection process to ensure transparency and reproducibility. What are the key components of the network architecture (e.g., number of layers, convolutional filters, activation functions)? How were suitable model parameters (e.g., learning rate, batch size, number of epochs) defined? Was hyperparameter tuning performed, and if so, what method was used (e.g., grid search, Bayesian optimization)?

4. The manuscript does not include an accuracy assessment of the FA maps generated by the trained deep-learning model. Without an independent validation dataset, it is unclear how reliable these maps are. The authors should conduct an accuracy assessment (e.g., confusion matrix, F1-score, precision-recall analysis) to quantify the model's performance.

5. The study integrates multiple datasets with different spatial resolutions, including MODIS (1 km), OLCI (300 m), and NOAA OISST SST (4 km). However, it does not explain how these resolution differences were addressed. Were resampling or aggregation techniques used to ensure consistency across datasets? If so, what methods were applied, and how might they have impacted the results? The manuscript should clarify the spatial integration process to improve reproducibility and reliability.

Overall, the study presents valuable findings on FA dynamics at a global scale, but key methodological and analytical concerns must be addressed to enhance the credibility and reproducibility of the results. Addressing the issues outlined above, particularly related to model validation, dataset integration, and comparative analyses, will significantly strengthen the manuscript.

Reviewer #2

(Remarks to the Author)

The scope of the submitted manuscript is to investigate the trend of the abundance and distribution of floating algal scums in the global oceans using satellite observations. The study focuses on both microalgae (phytoplankton) and macroalgae (seaweed). It provides a unique perspective on how the global oceanographic changes affect each species and how some of them are favored compared to others. A deep learning model is utilized to identify floating algal patches, whereas the species level classification is done through a spectral shape algorithm. The manuscript is on high level, well organized and structured and we believe that will add significant knowledge on the global oceans health.

- What are the noteworthy results?

This is a significant and valuable work that pinpoints a global scale implication of shifting climate and oceanographic parameters. The research results indicate that over the last decades, there is a significant and steady increase of both floating microalgae and macroalgae. Furthermore, from 2010 and onward, the rate of the abundance of macroalgae is significantly higher, which indicates that oceanographic parameter changes favor the growth of seaweed and that can lead to a regime shift. These findings identify a global trend and are not limited to a regional scale. Furthermore, the use of satellite data (MODIS and Sentinel-3 OLCI), ensures that this study can also be replicated in the future years and be directly comparable to these findings.

- Will the work be of significance to the field and related fields? How does it compare to the established literature? If the work is not original, please provide relevant references.

Yes, this work is a unique perspective on the distribution of floating algae in the global oceans and these changes are linked to the shifting climate and oceanographic conditions. To our knowledge, there are similar publications that discuss these issues, but they are limited either in scale (regional or on coastal environments) or in species (this work makes a distinction between species and does not reside only on proxies like chlorophyll-a).

- Does the work support the conclusions and claims, or is additional evidence needed?

The authors provide enough statistical measures to justify the positive trends of floating algal appearances. Furthermore, they provide extra data, like nutrient rations, dust depositions etc., to support their conclusions regarding the parameters that affect these positive trends that were found.

- Are there any flaws in the data analysis, interpretation, and conclusions? - Do these prohibit publication or require revision?

No, there are not. We did not find any statistical errors, and the results support the conclusions.

- Is the methodology sound? Does the work meet the expected standards in your field?

The methods used are already established, validated and published according to the authors. They provide the relative references within the manuscript (only one reference - Qi et al., 2023- is not in the reference list and should be added). As such, the results of known algorithms are scaled up at a global level.

- Is there enough detail provided in the methods for the work to be reproduced?

The methods in the current manuscript are not described in a way to make them reproducible. However, the authors provide a list of references that describe the methodology in details and the process they used to train and validate their models. Therefore, the methods of the current manuscript can be reproduced.

In conclusion, we believe that this manuscript would be a valuable addition to this journal and should be considered for publication. We propose some minor changes that should be addressed, that mostly refer to the conciseness of the manuscript as well as some formatting and referencing issues. Specifically:

1. Line 40: You need to add some supporting references to this statement.
2. Lines 43-44: Add 1-2 supporting references.
3. Lines 46-48: This sentence should probably be rephrased. It is a bit convoluted and the meaning is not clear enough.
4. Line 110: From this point on there are inconsistencies in the in-text formatting of the citations. The style changes from numbering to (name, year). Please change the text according to the appropriate style of the journal.
5. Line 110: This citation (Goodwin et al., 2022) is not listed in the bibliography section.
6. Line 147: Goes et al., 2018 is not listed in the bibliography section.
7. Lines 167-168: Can you clarify what you mean with this phrase? You integrated all microalgae species into one class and all macroalgae into another?
8. Line 183: Something is missing here. Probably you wanted to write "constrained by the".
9. Line 184-188: This section is not clear enough. We suggest you to rephrase it.
10. Line 184: You have already established that N stands for nitrogen in lines 52-53.
11. Line 184-188: In this paragraph/section you change between the use of "phosphorus" and "P", which is a bit confusing. You should decide into one use and stick to it through the manuscript.
12. Line 225: Change to "Yellow Sea (YS) and East China Sea (ECS)". You use these abbreviations later in the manuscript, but you do not define them.
13. There are two additional comments in the attached pdf file for the methods section which was not line numbered.

*Please see the attachment

Reviewer #3

(Remarks to the Author)

I co-reviewed this manuscript with one of the reviewers who provided the listed reports. This is part of the Nature

Communications initiative to facilitate training in peer review and to provide appropriate recognition for Early Career Researchers who co-review manuscripts.

Version 1:

Reviewer comments:

Reviewer #1

(Remarks to the Author)

The authors have thoroughly addressed the reviewers' comments through detailed revisions, enhancing the scientific rigor and readability of the paper. Key revisions include the model validation and dataset integration. These revisions effectively address the core issues raised in the initial review.

Reviewer #2

(Remarks to the Author)

All my remarks were addressed. I recommend this manuscript for publication.

Reviewer #3

(Remarks to the Author)

Summary of changes: We added more details to justify and clarify the selected methodology (Reviewer #1), and improved the presentation by addressing all editorial comments and suggestions (Reviewers #2 & #3).

REVIEWER COMMENTS

Reviewer #1 (Remarks to the Author):

I have completed the review of the manuscript titled "Global Floating Algae Are Rising," submitted for consideration in Nature Communications. This study presents a novel global analysis of floating algae (FA), using an extensive dataset of satellite imagery over a 20-year period. Including different species and associated environmental factors enhances its originality and significance. While the study contributes valuable insights into FA dynamics, certain issues must be addressed before the manuscript can be considered for publication. Below, I provide a detailed review of the article.

Reply: We thank the reviewer for recognizing the value of this work. Please see our response below.

1. The study employs a deep-learning model (ResUNet) for FA detection but does not provide sufficient details about the training and validation datasets.

The accuracy and reliability of a machine learning model depend heavily on the quality and quantity of these datasets. The manuscript states: "The ResUNet deep-learning model is trained and validated using carefully prepared 'ground truth' images, with detailed steps described in [48]. For simplicity, such details are not repeated here." However, reference [48] mentions that the ground-truth images were labeled using a combination of manual outlining and objective delineation. Given the vast number of MODIS images analyzed in this study, how was ground-truth data manually determined at such a large scale? Additionally, what quality control measures were implemented to ensure accuracy? The authors should provide a detailed explanation of the ground-truth dataset, including its accuracy, potential biases, and whether any independent validation was performed.

Reply: In the revision, we added more details on the model's validation across different FA types using independent "ground truth" images that were not used in model training. Although "manual" sounds tedious, in practice only the regions of interest (ROIs) were manually drawn in an image, while the image features within each ROI were delineated objectively using the local AFAI threshold. In total, 830 MODIS L2 "ground truth" images (6,912 sub-images with each sub-image being 256 x 256 pixels) covering the global oceans were prepared this way for model training and validation. Then the model was applied to all 1.2 million MODIS L2 images without any manual work. The uncertainties of the model were assessed using independent "ground truth" images, with statistical measures presented in Supplementary Materials.

For independent assessment of the model, we have uploaded all digital files, quick-look images, "ground truth" images, and model output images to a data repository (<https://doi.org/10.17632/f39zt9g2c4.1>) with detailed README files, which will all become available after approval by the data center (which usually takes 5-7 days). This way, others can test their models independently over the same dataset, and hopefully develop improved models in the future. This is now clarified in the Methods section.

2. The study employs the ResUNet deep-learning model but does not justify why this model was chosen over more advanced alternatives, such as Vision Transformers (ViT) or other state-of-the-art deep-learning architectures. A comparative analysis of model performance (e.g., accuracy, computational efficiency, robustness) would strengthen the manuscript's credibility. If ResUNet was selected due to specific advantages, these should be clearly outlined.

Reply: We thank the reviewer for raising this methodology question, and feel sorry for not making it clear in our original manuscript. As commented by Reviewer #2, the model is not new but has been established and used in several regional studies by the same authors and other researchers. It has been scaled up in this study after extensive training and validation before analyzing all floating algae (FA) types and their patterns on the global scale. As a matter of fact, an earlier version of the DL model has been used operationally in our Sargassum Watch System (SaWS, <https://optics.marine.usf.edu/projects/saws.html>) to generate macroalgae maps on a daily basis for the Atlantic Ocean, and these maps have been used widely by different users for a variety of purposes.

There is indeed more than one method to extract FA pixels from satellite imagery, for example by using a threshold-based approach (Qi et al., 2016; Wang and Hu, 2016) or other machine learning approaches (Wang and Hu, 2021). In the Supplemental Materials of Wang and Hu (2021), several machine learning approaches have been compared in their ability to extract *Sargassum* pixels, and these included UNet, Multi-Layer Perception, Random Forest, Support Vector Machine, and Boosted Regression Trees. UNet was found to have superior performance in model accuracy.

The ResUNet deep-model has shown superior performance over other approaches for this purpose (Hu et al., 2023a; Qi et al., 2023; Sun et al., 2024), where the principles and practices in detecting floating algae pixels and quantifying floating alga amount have been detailed in Hu et al. (2023b) and Hu et al. (2025). Therefore, because this paper is focused on the FA global patterns rather than on methodology development, we did not perform a comparison between the chosen method and other possible methods as the chosen method has already been established in the peer-reviewed literature.

Now recognizing the need for the completeness of the paper to a better readership, we added more details in the Methods section on the deep-learning model itself (see response to #3 below), and we also added a section (2.A) under Methods to explain why ResUNet deep-learning was selected among other possible choices, including ViT.

Briefly, while ViT or other transformers or other image segmentation techniques (e.g., discriminant superpixel graph or dSPG, Yu et al., 2024) have seen rapid increases in remote sensing, it has been primarily used in object recognition on land (see review by Aleissae et al., 2023) or water objects (e.g., oil slicks) clearly shown in meter-resolution images (Dong et al., 2023), while our target pixels are weak image features from water, with floating algae mostly occupying <1% of an image pixel (1-km resolution). These weak image features make them very difficult to extract as compared to features on land or in high-resolution images. In earlier trail-and-error tests, we did attempt to incorporate ViT's core component, namely "self attention", in the CNN-based ResUNet, but with unsatisfactory results: similar weak features due to cloud-adjacency effects, cloud shadows, or image noise were often falsely recognized as FA features. We therefore spent more time on optimizing the performance of ResUNet (see response to #3 below). This is now explained in the Methods section with Supplementary Figs. 1-4. Of course, this does not mean that our current ResUNet model is the best, but so far it is an established method for both floating macroalgae mats and microalgae scums, with accuracy of 90% or higher. In the future we will thoroughly evaluate its pros and cons in the context of other deep learning approaches, and hopefully make further improvements in model selection and model performance.

- Aleissae, A., et al. (2023). Transformers in remote sensing: A survey. *Remote Sensing*, 15(7), 1860. <https://doi.org/10.3390/rs15071860>
- Dong, X.; Li, J.; Li, B.; Jin, Y.; Miao, S. Marine Oil Spill Detection from Low-Quality SAR Remote Sensing Images. *J. Mar. Sci. Eng.* 2023, 11,1552. <https://doi.org/10.3390/jmse11081552>
- Hu, C., et al. (2023a). Mapping and quantifying pelagic Sargassum in the Atlantic Ocean using multi-band medium-resolution satellite data and deep learning. *Remote Sens. Environ.*, 113515, <https://doi.org/10.1016/j.rse.2023.113515>.
- Hu, C., et al. (2023b). Mapping Ulva prolifera green tides from space: A revisit on algorithm design and data products. *Int. J. Appl. Earth Obs. & Geoinfor.* 116, 103173, <https://doi.org/10.1016/j.jag.2022.103173>.
- Hu, C., et al. (2025). Monitoring pelagic Sargassum in the Atlantic Ocean from space: Principles and practices. *Harmful Algae*, 144, 102840, <https://doi.org/10.1016/j.hal.2025.102840>
- Sun, Y., et al. (2024). Continuous Sargassum monitoring across the Caribbean Sea and Central Atlantic using multi-sensor satellite observations. *Remote Sens. Environ.*, vol. 309, p. 114223, <https://doi.org/10.1016/j.rse.2024.114223>
- Qi, L., et al. (2016). Long-term trend of Ulva prolifera blooms in the western Yellow Sea. *Harmful Algae*, 58:35-44. <http://dx.doi.org/10.1016/j.hal.2016.07.004>
- Qi, L., M. Wang, C. Hu, D. G. Capone, A. Subramaniam, E. J. Carpenter, and Y. Xie, (2023). Trichodesmium around Australia: A view from space. *Geophysical Research Letters*, 50, e2023GL104092. <https://doi.org/10.1029/2023GL104092>.
- Wang, M., and C. Hu (2016). Mapping and quantifying Sargassum distribution and coverage in the Central West Atlantic using MODIS observations. *Remote Sens. Environ.*, 183:356-367. <http://dx.doi.org/10.1016/j.rse.2016.04.019>.
- Wang, M., and C. Hu (2021). Satellite remote sensing of pelagic Sargassum macroalgae: The power of high resolution and deep learning. *Remote Sens. Environ.*, 264, 112631, <https://doi.org/10.1016/j.rse.2021.112631>
- Yu, L, et al. (2024). dSPG: A New Discriminant Superpixel Graph Regularizer and Convolutional Network for Hyperspectral Image Classification. *IEEE Transactions on Geoscience and Remote Sensing*, 62, 5526118, doi: 10.1109/TGRS.2024.3439434.

3. The manuscript does not provide sufficient details on the ResUNet model's structure. The authors should clearly describe the model architecture and parameter selection process to ensure transparency and reproducibility. What are the key components of the network architecture (e.g., number of layers, convolutional filters, activation functions)? How were suitable model parameters (e.g., learning rate, batch size, number of epochs) defined? Was hyperparameter tuning performed, and if so, what method was used (e.g., grid search, Bayesian optimization)?

Reply: We thank the reviewer for reminding this. Such model details have been presented in our earlier publications (e.g., Supplemental Materials in Hu et al., 2023, where model structure and optimization details are presented). In the revision we have added model details in the Methods section, together with Supplementary Figs. 1-4 to clarify the details.

4. The manuscript does not include an accuracy assessment of the FA maps generated by the trained deep-learning model. Without an independent validation dataset, it is unclear how reliable these maps are. The authors should conduct an accuracy assessment (e.g., confusion matrix, F1-score, precision-recall analysis) to quantify the model's performance.

Reply: We have added accuracy assessment for both macroalgae and microalgae in the Methods section, with results presented in Supplementary Table 1. These are similar to previously published accuracy assessment for regional studies (e.g., F1-score ~ 0.9) but at a much broader (i.e., global) scale than any previous publications.

5. The study integrates multiple datasets with different spatial resolutions, including MODIS (1 km), OLCI (300 m), and NOAA OISST SST (4 km). However, it does not explain how these resolution differences were addressed. Were

resampling or aggregation techniques used to ensure consistency across datasets? If so, what methods were applied, and how might they have impacted the results? The manuscript should clarify the spatial integration process to improve reproducibility and reliability.

Reply: these datasets were used separately for different purposes: MODIS (1km) for global mapping, OCLI (300 m) for spectral diagnostics, and OISST (4 km) for studying environmental conditions. Most variables were interpolated to the MODIS resolution but OLCI was retained at 300 m resolution to spectrally diagnose the image features corresponding to those found in the MODIS images. We clarified this in the Methods section.

Overall, the study presents valuable findings on FA dynamics at a global scale, but key methodological and analytical concerns must be addressed to enhance the credibility and reproducibility of the results. Addressing the issues outlined above, particularly related to model validation, dataset integration, and comparative analyses, will significantly strengthen the manuscript.

Reply: We thank the reviewer for this positive feedback. Although the deep-learning model has been published in several regional studies with model details and performance measures (as well as comparison with other machine learning approaches) presented, we tried our best to justify its use and provided more details on the model structure and accuracy in the revised manuscript. We agree that this addition will strengthen the manuscript by making it more complete. We have also uploaded all required data files to a data repository (<https://doi.org/10.17632/f39zt9g2c4.1>) for others to test and verify, and hopefully will make further improvements in model selection and model performance through community efforts.

Reviewer #2 (Remarks to the Author):

The scope of the submitted manuscript is to investigate the trend of the abundance and distribution of floating algal scums in the global oceans using satellite observations. The study focuses on both microalgae (phytoplankton) and macroalgae (seaweed). It provides a unique perspective on how the global oceanographic changes affect each species and how some of them are favored compared to others. A deep learning model is utilized to identify floating algal patches, whereas the species level classification is done through a spectral shape algorithm. The manuscript is on high level, well organized and structured and we believe that will add significant knowledge on the global oceans health.

Reply: We thank the reviewer for these encouraging comments.

- What are the noteworthy results?

This is a significant and valuable work that pinpoints a global scale implication of shifting climate and oceanographic parameters. The research results indicate that over the last decades, there is a significant and steady increase of both floating microalgae and macroalgae. Furthermore, from 2010 and onward, the rate of the abundance of macroalgae is significantly higher, which indicates that oceanographic parameter changes favor the growth of seaweed and that can lead to a regime shift. These findings identify a global trend and are not limited to a regional scale. Furthermore, the use of satellite data (MODIS and Sentinel-3 OLCI), ensures that this study can also be replicated in the future years and be directly comparable to these findings.

Reply: We thank the reviewer for the positive feedback.

- Will the work be of significance to the field and related fields? How does it compare to the established literature? If the work is not original, please provide relevant references.

Yes, this work is a unique perspective on the distribution of floating algae in the global oceans and these changes are linked to the shifting climate and oceanographic conditions. To our knowledge, there are similar publications that discuss these issues, but they are limited either in scale (regional or on coastal environments) or in species (this work makes a distinction between species and does not reside only on proxies like chlorophyll-a).

Reply: We thank the reviewer for recognizing the value of this work.

- Does the work support the conclusions and claims, or is additional evidence needed?

The authors provide enough statistical measures to justify the positive trends of floating algal appearances. Furthermore, they provide extra data, like nutrient rations, dust depositions etc., to support their conclusions regarding the parameters that affect these positive trends that were found.

Reply: We thank the reviewer for agreeing with our statistics and findings.

- Are there any flaws in the data analysis, interpretation, and conclusions? - Do these prohibit publication or require revision?

No, there are not. We did not find any statistical errors, and the results support the conclusions.

Reply: We thank the reviewer for the positive feedback.

- Is the methodology sound? Does the work meet the expected standards in your field?

The methods used are already established, validated and published according to the authors. They provide the relative references within the manuscript (only one reference - Qi et al., 2023- is not in the reference list and should be added). As such, the results of known algorithms are scaled up at a global level.

Reply: We thank the reviewer for capturing this point.

- Is there enough detail provided in the methods for the work to be reproduced?

The methods in the current manuscript are not described in a way to make them reproducible. However, the authors provide a list of references that describe the methodology in details and the process they used to train and validate their models. Therefore, the methods of the current manuscript can be reproduced.

Reply: We thank the reviewer for capturing this point. Per Reviewer #1's comments on methodology, however, we now included more descriptions and validations of the method in Supplemental materials.

In conclusion, we believe that this manuscript would be a valuable addition to this journal and should be considered for publication. We propose some minor changes that should be addressed, that mostly refer to the conciseness of the manuscript as well as some formatting and referencing issues.

Reply: We appreciate the reviewer's time in reading this manuscript carefully and effort in helping to improve its presentation for a better readership.

Specifically:

1. Line 40: You need to add some supporting references to this statement.

Reply: we have added several relevant references to support this statement.

2. Lines 43-44: Add 1-2 supporting references.

3. Lines 46-48: This sentence should probably be rephrased. It is a bit convoluted and the meaning is not clear enough.

4. Line 110: From this point on there are inconsistencies in the in-text formatting of the citations. The style changes from numbering to (name, year). Please change the text according to the appropriate style of the journal.
5. Line 110: This citation (Goodwin et al., 2022) is not listed in the bibliography section.
6. Line 147: Goes et al., 2018 is not listed in the bibliography section.
7. Lines 167-168: Can you clarify what you mean with this phrase? You integrated all microalgae species into one class and all macroalgae into another?
8. Line 183: Something is missing here. Probably you wanted to write "constrained by the".
9. Line 184-188: This section is not clear enough. We suggest you to rephrase it.
10. Line 184: You have already established that N stands for nitrogen in lines 52-53.
11. Line 184-188: In this paragraph/section you change between the use of "phosphorus" and "P", which is a bit confusing. You should decide into one use and stick to it through the manuscript.
12. Line 225: Change to "Yellow Sea (YS) and East China Sea (ECS)". You use these abbreviations later in the manuscript, but you do not define them.
13. There are two additional comments in the attached pdf file for the methods section which was not line numbered.

*Please see the attachment

Reply: all annotated editorial comments above are addressed in the revised manuscript. We thank the reviewer for these careful thoughts.

Reviewer #3 (Remarks to the Author):

Reply: We appreciate the reviewer's effort in helping to improve the presentation of this work.